# Using DEFORM Software for Determination of Parameters for Two Fracture Criteria on DIN 34CrNiMo6

**Ivana Poláková \*, Michal Zemko, Martin Rund and Ján Džugan** 

COMTES FHT, Průmyslová 995, Dobřany 334 41, Czech Republic; mzemko@comtesfht.cz (M.Z.);
mrund@comtesfht.cz (M.R.); jdzugan@comtesfht.cz (J.D.)

\* Correspondence: ivana.polakova@comtesfht.cz; Tel.: +420-377-197-321

**Abstract:** The aim of this study was to calibrate a material model with two fracture criteria that is available in the DEFORM software on DIN 34CrNiMo6. The purpose is to propose a type of simple test that will be sufficient for the determination of damage parameters. The influence of the quantity of mechanical tests on the accuracy of the fracture criterion was explored. This approach was validated using several tests and simulations of damage in a tube and a round bar. These tests are used in engineering applications for their ease of manufacturing and their strong ability to fracture. The prediction of the time and location of the failure was based on the parameters of the relevant damage model. Normalized Cockroft-Latham and Oyane criteria were explored. The validation involved comparing the results of numerical simulation against the test data. The accuracy of prediction of fracture for various stress states using the criteria was evaluated. Both fracture criteria showed good agreement in terms of the fracture locus, but the Oyane criterion proved more suitable for cases covering larger triaxiality ranges.

**Keywords:** flow stress; stress triaxiality; material damage; FEM simulation

## 1. Introduction

The aim of this paper is to develop a procedure for failure prediction through simulations when a limited amount of tests are available. The purpose of the paper is to demonstrate quick and sufficiently accurate predictions based on very few—and if possible simple—experiments. From engineering and practical points of view, the number of tests on specimens should be no higher than absolutely necessary. When it comes to calibrating fracture criteria, the challenge lies in choosing the right tests [1]. DEFORM software (developed by Scientific Forming Technologies Corporation, Columbus, OH, USA) was used for all computational modelling tasks presented in this paper. Two fracture criteria available in DEFORM were chosen for validating the procedure of failure prediction: the Normalized Cockcroft-Latham criterion (NCL) and the Oyane criterion.

An accurate description of material behaviour is a prerequisite for successful modelling of cold metal forming processes. Strain hardening (flow stress) and fracture behaviour depend on basic material properties, which are governed by processing history, typically the temperature and strain rate profiles. Such properties can be determined through experiments and testing. In forming processes, where fracture must be avoided, identification of forming limits is a matter of key importance. However, the ability to determine the conditions that lead to fracture is important for processes in which it is undesirable, as well as in processes where material separation is the desired outcome (such as cutting).

Numerous researchers have attempted to predict damage and fracture [2–4] and a multitude of models have been proposed over the years. These include Bridgman [5], McClintock [6] and Rice and

Tracey [7], who demonstrated the effect of hydrostatic pressure on fracture. Wilkins et al. postulated that both hydrostatic pressure and deviatoric stress contribute to damage. Johnson and Cook [8] introduced a fracture model that accounted for strain rate and temperature.

Bao and Wierzbicki [9] performed numerous experiments involving a wide range of triaxialities and suggested that the dependence of fracture strain on stress triaxiality was not a monotonic curve. They claimed that there were different fracture mechanisms operating in different stress triaxiality ranges. Later, these authors proposed a cut-off value for triaxiality at negative 1/3, where fracture strain becomes infinity. Kim et al. [10], Gao et al. [11], and Brünig et al. [12] studied triaxiality and Lode angle. This has been followed up by many authors, including our laboratory, to develop testing procedures and the shape of the samples [13–15].

The failure prediction procedure described in this paper consists of defining material behaviour for numerical simulation on the basis of several different tests and finding correlation between them and the accuracy and usability of the prediction. For this purpose, mechanical tests classified into three groups were performed. The first one involved tensile and compression tests through which flow stress data were acquired. The second group comprised notched tensile test, shear test and plane strain test conducted to determine fracture parameters. The last group of tests were performed to validate the fracture criteria. These included radial compression of a ring specimen and axial compression of a notched cylinder.

## 2. Material

Fracture modelling for tests was carried out on DIN 34CrNiMo6 steel at room temperature and at quasi-static strain rates. This steel was chosen due to its versatile engineering applications [16]. It is used for heavy-duty components, such as axles, shafts, crankshafts, connecting rods, valves, propeller hubs, gears, couplings, torsion bars, aircraft components, and heavy parts of rock drills.

The advantages of DIN 34CrNiMo6 martensitic steel include high ductility, hardenability and strength [16]. In fact, it provides the highest yield strength and ultimate tensile strength from the entire class of high strength steels. Generally, martensitic steels are characterized by a martensitic matrix produced by quenching with small amounts of ferrite or bainite. This steel is strengthened by the precipitation of a fine dispersion of carbides during tempering. Its nominal chemical composition, in weight percent, is presented in Table 1. The material was heat-treated to 34 HRC. Basic measured mechanical data are shown in Table 2.

**Table 1.** Nominal chemical composition of DIN 34CrNiMo6.

| Element | C | Si | Mn | Cr | Mo | Ni |
|---|---|---|---|---|---|---|
| Weight % | 0.34 | ≤0.40 | 0.65 | 1.50 | 0.22 | 1.50 |

**Table 2.** Measured mechanical data of DIN 34CrNiMo6.

| Yield Strength (MPa) | Tensile Strength (MPa) | Elongation (%) |
|---|---|---|
| 877 | 990 | 6.6 |

## 3. Uncoupled Fracture Models

Fracture models are divided into coupled and uncoupled categories. Coupled models are complex, require a great many material constants, and are computationally demanding. In uncoupled models, the plasticity model is separate from the failure model. This means a significant reduction in the computation effort. The DEFORM software only offers uncoupled fracture criteria.

Two fracture criteria were selected for this paper. Their applicability to industrial problems was evaluated. The NCL criterion has only one parameter which governs the critical damage. Therefore, the nature of the problem at hand needs to be accounted for. Identifying the applicability range of a criterion is an essential task. Nevertheless, the NCL criterion is based on maximal principal stress,

which is dependent on both the triaxiality and the Lode angle. The NCL criterion only averages the effect of both parameters using maximal principal stress. Generally, fracture criteria with multiple parameters are applicable to a wider range of stress states. One of them is the Oyane criterion, the other criterion explored in this work. The Oyane criterion only includes stress triaxiality dependence. Oyane prefers triaxiality, which seems to be more significant for crack initiation and evolution.

The stress state in a material is affected by external loads. Tensile stress leads to damage more rapidly than compression. Physical damage is activated by microvoid nucleation and propagates through their growth and coalescence. The development of damage is studied in correlation to stress triaxiality. Stress triaxiality is defined as a ratio of the mean hydrostatic stress $\sigma_m$ to the equivalent stress $\sigma_{eq}$:

$$\eta = \frac{\sigma_m}{\sigma_{eq}} = \frac{\sigma_1 + \sigma_2 + \sigma_3}{3\sigma_{eq}}, \tag{1}$$

where $\sigma_i$ ($i$ = 1, 2, 3) denotes the principal stress. The equivalent stress $\sigma_{eq}$ is defined in terms of three principal stresses. Assuming that the material is isotropic, the equivalent stress is given by the von Mises equation:

$$\sigma_{eq} = \frac{1}{\sqrt{2}} \sqrt{\left[ (\sigma_1 - \sigma_2)^2 + (\sigma_2 - \sigma_3)^2 + (\sigma_3 - \sigma_1)^2 \right]}, \tag{2}$$

The equivalent strain $\varepsilon_{eq}$ is expressed as:

$$\varepsilon_{eq} = \frac{\sqrt{2}}{3} \sqrt{\left[ (\varepsilon_1 - \varepsilon_2)^2 + (\varepsilon_2 - \varepsilon_3)^2 + (\varepsilon_3 - \varepsilon_1)^2 \right]}, \tag{3}$$

Many published experimental reports [17] have shown that fracture strain decreases with increasing triaxiality, although the dependence is not monotonic as in Figure 1. A peak is normally found near the 1/3 value, which corresponds to a smooth tensile test. Another factor in the occurrence of fracture is the Lode parameter. This is related to the third deviatoric invariant and reflects shear deformation. The value of the Lode parameter is 0 for plane strain conditions, +1 for axisymmetric tension, and −1 for axisymmetric compression. Data from various experimental tests plotted in a fracture locus graph provide the basis for fracture prediction, see example in Figure 1, where the dependence of the fracture strain on stress triaxiality and the Lode parameter is plotted. The black line in the graph correspond to plane stress. The critical level of accumulated damage is reached when equivalent strain exceeds a limit value.

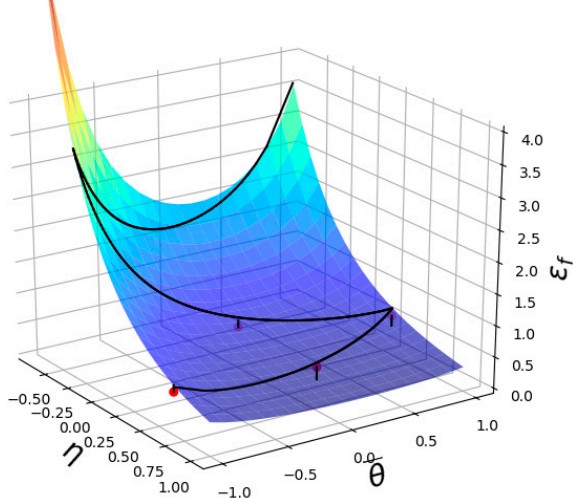

**Figure 1.** Representation of fracture locus in space.

In computer simulation, the critical value of a damage criterion indicates the initiation of fracture. The actual fracture strain can be determined from a simulation of the real test. In DEFORM code, the growth and coalescence of voids that lead to the evolution of fracture can be modelled by either the softening method or by element deletion. The critical damage value depends on the stress state in the test and on the fracture model used.

In the NCL criterion, the largest principal stress is the most relevant factor for the initiation of fracture:

$$D_C = \int_0^{\varepsilon_f} \frac{\sigma_1}{\sigma_{eq}} d\varepsilon \,,$$  (4)

where $D_c$ denotes the critical damage value, $\varepsilon_f$ is the fracture strain, $\sigma_1$ stands for the maximal principal stress and $\sigma_{eq}$ is the equivalent stress. The maximal principal stress can be expressed by stress triaxiality and Lode angle [18,19]:

$$\sigma_1 = \sigma_{eq}\left(\eta + \frac{2}{3}cos\frac{\pi}{6}\left(1 - \overline{\theta}\right)\right),$$  (5)

where $\eta$ is triaxiality and $\overline{\theta}$ is a normalized Lode angle.

The Oyane fracture criterion is based on stress triaxiality as the main reason for fracture:

$$D_C = \int_0^{\varepsilon_f}\left[1 + \frac{1}{a_0}\frac{\sigma_m}{\sigma_{eq}}\right]d\varepsilon,$$  (6)

where $a_0$ is a material coefficient to be determined experimentally and the ratio $\sigma_m/\sigma_{eq}$ denotes the stress triaxiality.

The dimensionless normalized Lode angle [17] is defined in terms of invariants of the stress tensor:

$$I_1 = p = -\sigma_m = -\frac{1}{3}(\sigma_1 + \sigma_2 + \sigma_3),$$  (7)

$$I_2 = q = \sigma_{eq} = \sqrt{\frac{1}{2}\left[(\sigma_1 - \sigma_2)^2 + (\sigma_2 - \sigma_3)^2 + (\sigma_3 - \sigma_1)^2\right]},$$  (8)

$$I_3 = r = \frac{27}{2}\left[(\sigma_1 - \sigma_m)(\sigma_2 - \sigma_m)(\sigma_3 - \sigma_m)\right]^{\frac{1}{3}},$$  (9)

$$\overline{\theta} = 1 - \frac{2}{\pi}arccos\left(\left(\frac{r}{q}\right)^3\right),$$  (10)

where $I_i$ ($i$ = 1, 2, 3) are invariants of the stress tensor and $\overline{\theta}$ is the normalized Lode angle parameter. Only the NCL criterion has a built-in dependence on the Lode angle. The effect of the Lode angle on Oyane criterion can be done by separating the fracture coefficients for tests with different Lode angle.

## 4. Experiments

In the following sections, three groups of tests are described. The basic tests include tensile and compression tests. Tests for fracture coefficients determination comprised a tensile test of a notched specimen, plane strain, and shear tests. Validation tests involved notched and unnotched rings and notched cylinder specimens.

### 4.1. Basic Tests

Tensile testing was carried out at room temperature. The test data were used for flow stress calibration. A drawing of the smooth tensile specimen is shown in Figure 2; the dimensions are in millimetres. The figure also shows the locations and the distance of the landmarks (red crosses) monitored by an optical (laser) extensometer. The rate of loading for smooth tensile specimens was 2 mm/min. The initial strain rate was 0.001 s$^{-1}$. Elongation was measured by optical and mechanical extensometers but only the optical extensometer data were used for calculations.

The other tests described below were recorded using a DIC (Digital Image Correlation) system, except for the tensile tests of smooth and notched specimens, which were measured by an optical extensometer. A stochastic pattern was applied by airbrush on samples recorded with the DIC system. The actual plastic strain on the specimen surface was calculated using a correlation algorithm implemented in GOM Aramis software [20]. The resolution of cameras was 4248 × 2832 pixels. The frame rate was 5 Hz.

Cylindrical specimens 10 mm in diameter and 15 mm in height were used for compression testing. The locations of and the distance between the landmarks for load-displacement measurement are indicated with red crosses as indicated in Figure 3. The displacement was measured using DIC. The camera was pointed at the specimen and dies, onto which a pattern was applied by spraying to provide contrast for strain measurement and for calculation by means of GOM Aramis software.

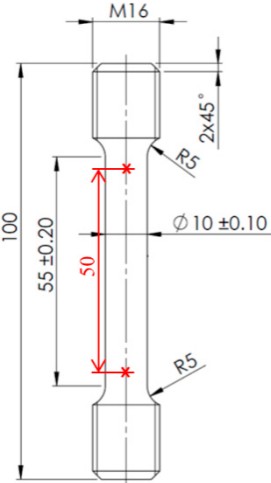

**Figure 2.** Drawing of tensile specimen.

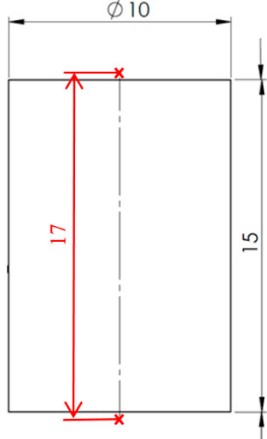

**Figure 3.** Drawing of compression specimen. The distance shown in red refers to the dies.

*4.2. Tests for Fracture Coefficient Determination*

The critical value of damage is estimated from plastic strain accumulated prior to fracture, which depends on the stress state. The choice of these tests for calibrating the fracture criteria was intended to cover a wide range of triaxialities and was based on literature survey and previous experience of the authors. They included the notched tensile test (Figure 4), shear test (Figure 5), and plane strain test (Figure 6).

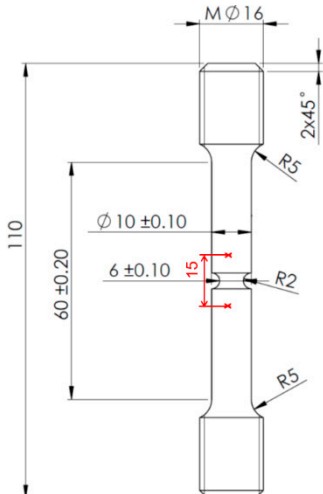

**Figure 4.** Drawing of notched tensile test specimen.

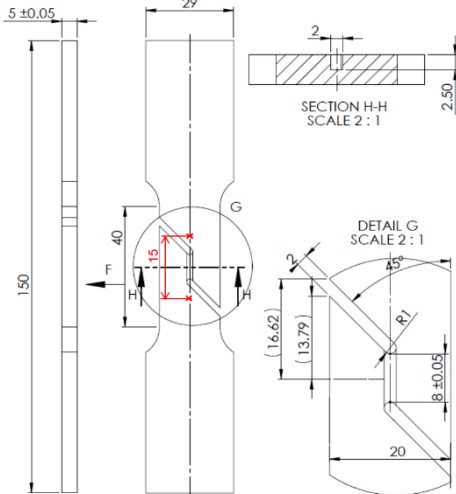

**Figure 5.** Drawing of shear test specimen.

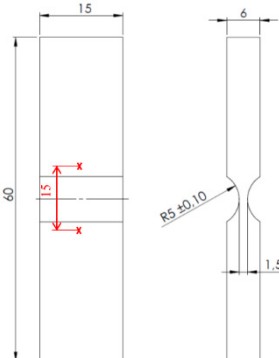

**Figure 6.** Drawing of plane strain test specimen.

Tensile testing of notched specimens at a loading velocity of 0.25 mm/min produced the same strain rate as in the previous tests. Optical and mechanical extensometers were used for measuring the gauge length by the red crosses as indicated in Figure 4. Fractured tensile specimens after testing are shown in Figure 7.

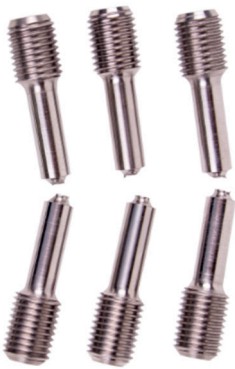

**Figure 7.** Notched tensile test specimens.

The purpose of the shear test and plane strain test was to cover a wider range of stress states for this study. The loading velocity was set at 0.5 mm/min in order to obtain the same strain rate in the relevant region as in notched tensile test. In both tests, extension was measured by DIC. Drawings of shear test and plane strain test specimens are presented in Figures 5 and 6, respectively. The gauge lengths are indicated by red crosses. Some fractured specimens with their surface patterns are shown in Figures 8 and 9.

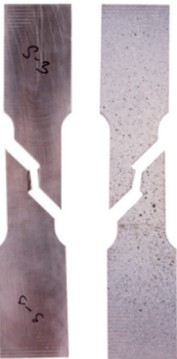

**Figure 8.** Shear test specimens.



**Figure 9.** Plane strain test specimens.

*4.3. Validation Tests*

The proposed fracture criteria parameters were validated using two simple tests. One involved radial compression of notched and unnotched rings, see Figures 10 and 11. The other was axial compression of a notched cylinder, as shown in Figure 12.

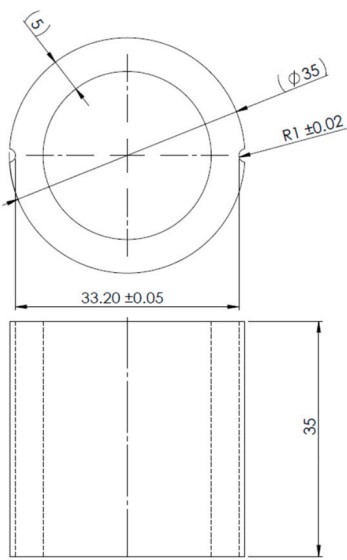

**Figure 10.** Drawing of notched ring specimen.

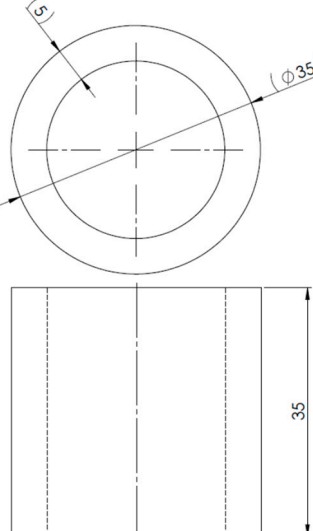

**Figure 11.** Drawing of unnotched ring specimen.

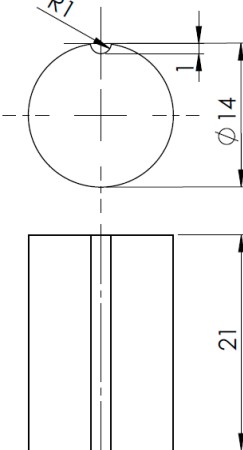

**Figure 12.** Drawing of the notched cylinder.

Based on an extensive literature survey, it was found that the presented geometry of test specimens is easy to manufacture and has a strong capacity for fracture initiation. These tests are not standardized, and are used for engineering purposes, such as assessing the damage mechanisms in pipes with and without defect, or the load response of a round bar with a defect.

A loading velocity of 2 mm/min was applied at room temperature. A DIC system was used for measuring the specimen displacement. Photographs of notched and unnotched specimens after testing are shown in Figure 13. Notched cylinders after axial compression are displayed in Figure 14. The cracked specimens clearly demonstrate that the failure was a shear-dominated process in this case.

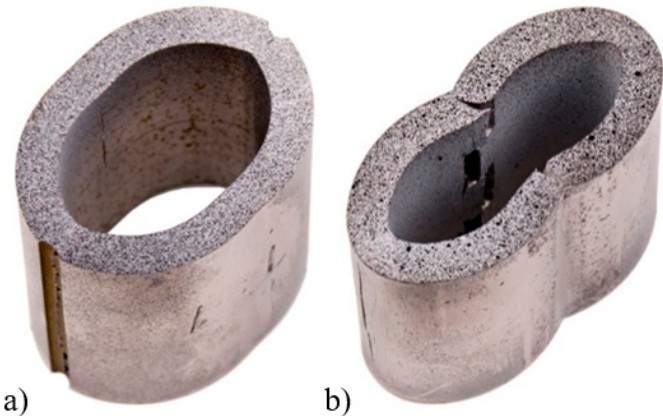

**Figure 13.** (**a**) Notched and (**b**) unnotched rings after testing.

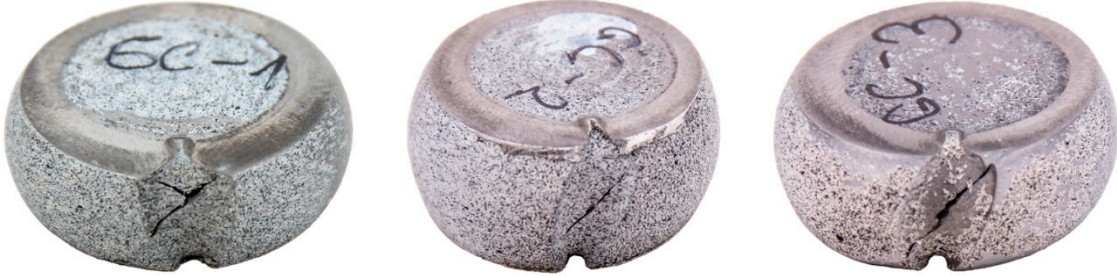

**Figure 14.** Notched cylinder specimens after testing with fractures in the notch region.

## 5. Numerical Simulation Using DEFORM Software

### 5.1. Construction of Plasticity Models

Stress–strain (flow stress) curves for the material were calibrated against data from the smooth tensile and compression tests. Obtaining flow stress curves from uniaxial tensile test data up to the onset of necking is relatively straightforward. This technique has been reported by several authors [21]. However, a disadvantage of tensile testing is that its flow stress data applies at very small strains only.

Much higher strain values are attained in compression than in tension. In compression tests, friction can be a significant issue. It should be reduced as much as possible to prevent specimen barrelling. Using compression test data to expand the flow stress range obtained from tensile tests is a common technique. The resulting flow stress curve is then more accurate at larger strains. The complete set of flow stress data used for simulations in this study is summarized in Table 3 and plotted in Figure 15.

**Table 3.** Flow stress data in a tabular form.

| Strain (-) | Flow Stress (MPa) | Strain (-) | Flow Stress (MPa) |
|---|---|---|---|
| 0 | 978 | 0.2 | 1210 |
| 0.005 | 1004 | 0.3 | 1230 |
| 0.01 | 1025 | 0.4 | 1242 |
| 0.015 | 1046 | 0.5 | 1255 |
| 0.02 | 1068 | 0.6 | 1268 |
| 0.03 | 1100 | 0.7 | 1282 |
| 0.04 | 1122 | 0.8 | 1295 |
| 0.05 | 1135 | 1 | 1325 |
| 0.06 | 1150 | 1.15 | 1348 |
| 0.1 | 1180 | 1.2 | 1356 |

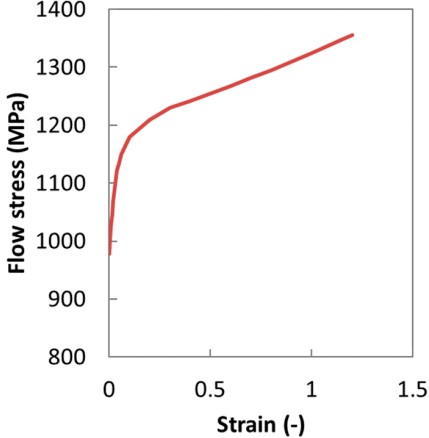

**Figure 15.** Flow stress plot.

Using the flow stress data calculated from tensile and compressive test data, these tests were simulated, replicating the real test conditions. Force-extension diagrams with the actually measured curves and the curves obtained by simulation using fitted flow stress data are shown in Figures 16 and 17 for the smooth tensile test and the compression test, respectively. Both tests were modelled as axisymmetric problems. A fine mesh with elements of 0.25 mm in size was used in the region where fracture was expected to occur. The reason was that strain gradient becomes steep in this region. Remeshing was applied when the specimen shape changed.

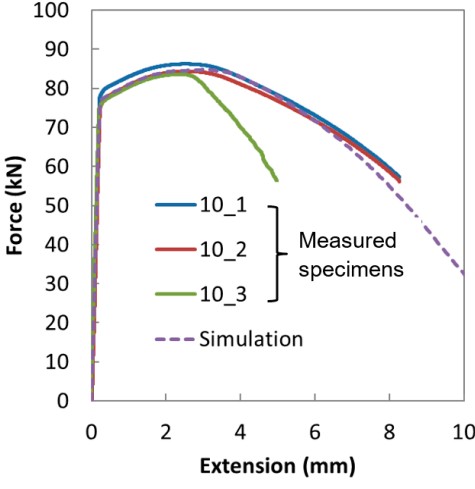

**Figure 16.** Comparison of tensile test curves.

None of the smooth tensile specimens fractured at mid-length. Specimens 10_1 and 10_2 exhibited almost identical trends, see Figure 16. Specimen 10_3 fractured earlier than the other two. All the measured compression test curves were almost identical. The simulated load response is in very good agreement with the measured one. The aim of these simulations was to verify the plasticity data to be used for modelling. No fracture criteria were considered at this stage.

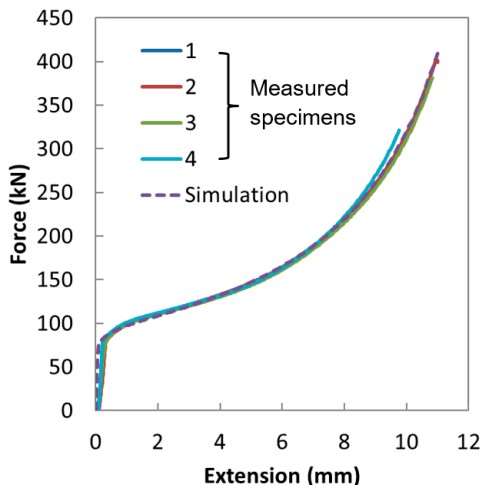

**Figure 17.** Comparison of compression test curves.

## 5.2. Identification of Fracture Parameters

To determine the critical damage threshold for simulation, experimental fracture data are needed [22]. Fracture is assumed to occur when the force-displacement curve begins to drop. This indicates fracture initiation and the amount of fracture strain. However, since the specimen deforms rapidly during necking, the exact fracture strain value it is very difficult to identify from measured test plots. An estimate of fracture strain can also be obtained by measuring the cross-sectional area of a specimen before and after testing [23]. Using FE simulation of tests, the onset of fracture and the fracture strain can be determined more accurately.

The measured and simulated forces are compared in Figure 18 for the notched tensile test, in Figure 19 for the shear test, and in Figure 20. for the plane strain test. These simulations were based on flow stress data from the previous tests. The measured and simulated forces are in very good agreement for all the tests, which means that the flow stress data from smooth tensile and compression tests had been determined accurately. The conditions and constraints used in the simulations of notched tensile test, shear and plane strain tests replicated the actual test conditions. A fine mesh with elements 0.15–0.3 mm in size was used at the mid-length region of the specimen.

The amount of fracture strain in the simulated notched tensile test can be found at the notch tip where failure was expected to occur. The point is indicated by an arrow in Figure 21. In the shear test simulation, a point on the specimen surface was chosen for this purpose where strain reaches the highest values, as in Figure 22. The location of the highest strain in the plane strain specimen is not on the surface but inside the specimen, see Figure 23. The moment when the crack is detected by DIC on the surface of the plane strain specimen may be different. The average strain on the cross-section of the specimen can be smaller than the fracture strain.

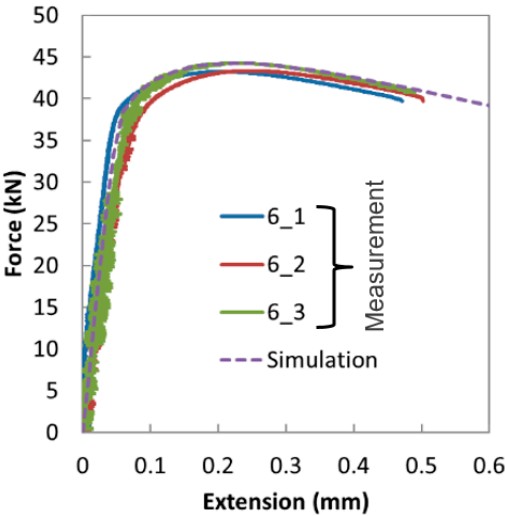

**Figure 18.** Comparison of forces for notched tensile test.

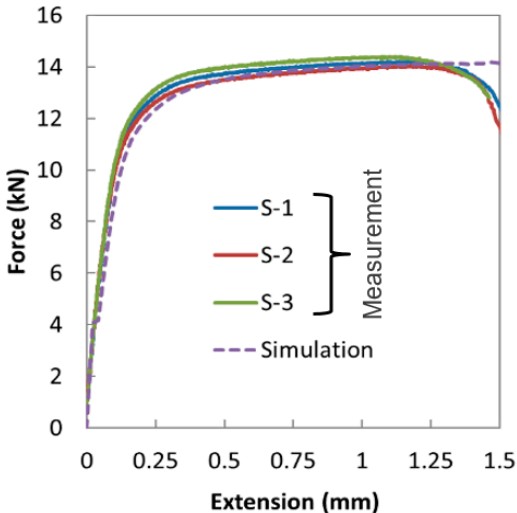

**Figure 19.** Comparison of forces for shear test.

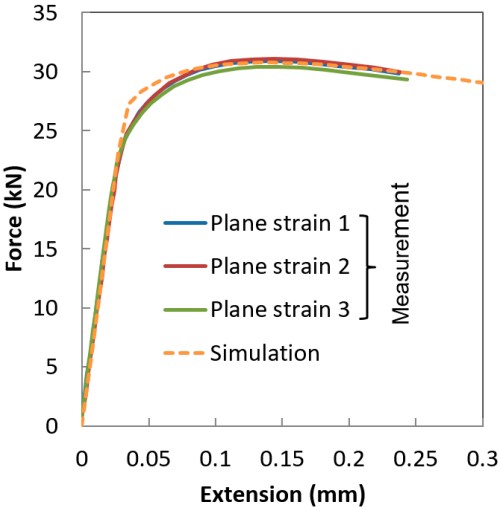

**Figure 20.** Comparison of forces for plane strain test.

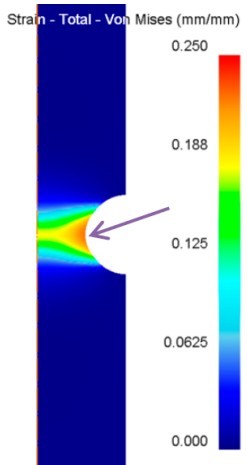

**Figure 21.** The fracture location in 2D simulation of notched tensile test.

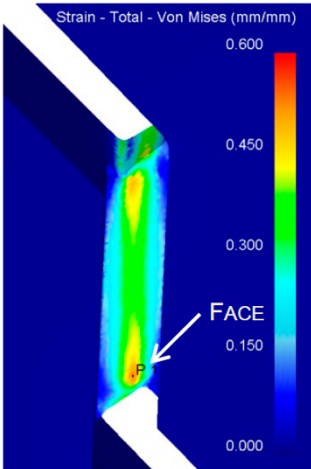

**Figure 22.** The fracture point in simulation of the shear test.

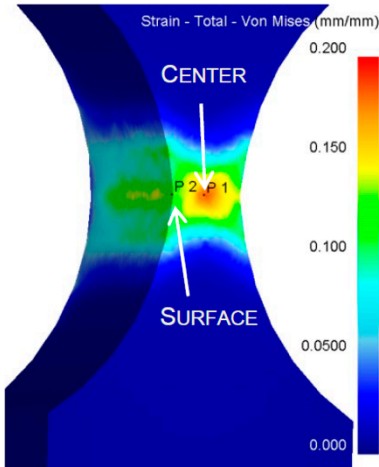

**Figure 23.** The fracture point in simulation of the plane strain test (longitudinal section).

Fracture strain at crack initiation, as determined from simulations calibrated against the actual test data, are listed in Table 4. Cracking was considered to initiate when the measured force curve either dropped off or began to deviate from the simulated force curve. This time instant and location indicate the critical damage value for the NCL criterion which is characterized by normalized Lode angle and

triaxiality. The parameters can be obtained directly in the DEFORM simulation data. The critical damage value is identical, accidentally, for the notched tensile specimen and the plane strain specimen.

**Table 4.** Critical damage values identified by simulation.

| Specimen Type | Fracture Strain (-) | NCL Critical Damage (-) | Average Triaxiality (-) | Normalized Lode Angle (-) |
|---|---|---|---|---|
| Notched tensile specimen | 0.22 | 0.23 | 0.40 | 0.91 |
| Shear test specimen | 0.55 | 0.38 | 0.06 | 0.16 |
| Plane strain specimen | 0.18 | 0.23 | 0.71 | 0.00 |

The calculated triaxiality paths at selected points on the specimens are plotted in Figures 24–26. The triaxiality is changing significantly during the stress evolution, so an average value is usually used and recommended. The average value of triaxiality $\eta_{av}$ is calculated based on the following equation:

$$\eta_{av} = \int_0^{\varepsilon_f} \frac{\eta}{\varepsilon_f} d\varepsilon . \tag{11}$$

The normalized Lode angle paths at identical points in test specimens are plotted in Figures 27–29. The normalized Lode angle for the notched tensile test is close to 1. In the shear test, the normalized Lode angle is close to 0, similar to in the plane strain test. These and the average stress triaxiality values indicate that the choice of these tests was appropriate to reflect the different stress states in the material. All average triaxiality and Lode angle values are given in Table 2.

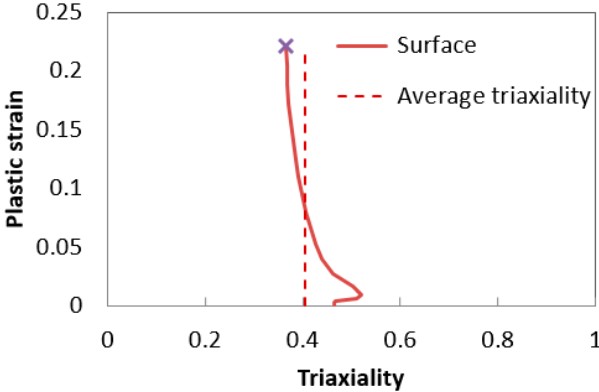

**Figure 24.** Triaxiality path for notched tensile test.

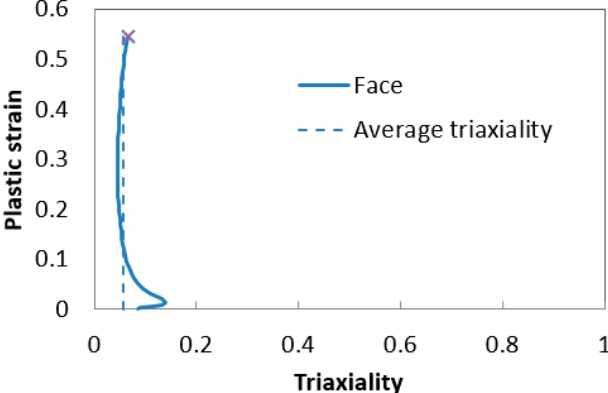

**Figure 25.** Triaxiality path for shear test.

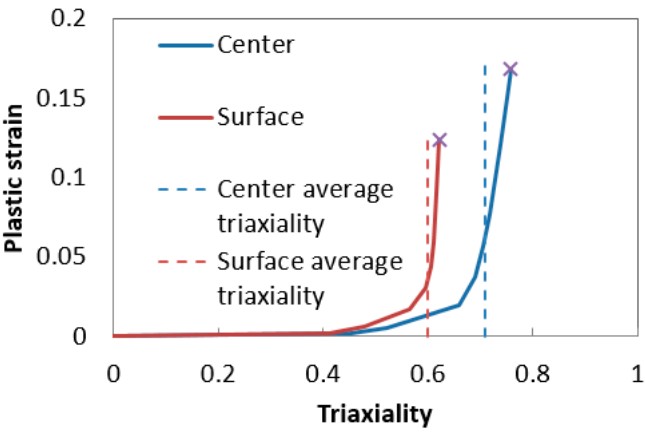

**Figure 26.** Triaxiality paths for the plane strain test at two points.

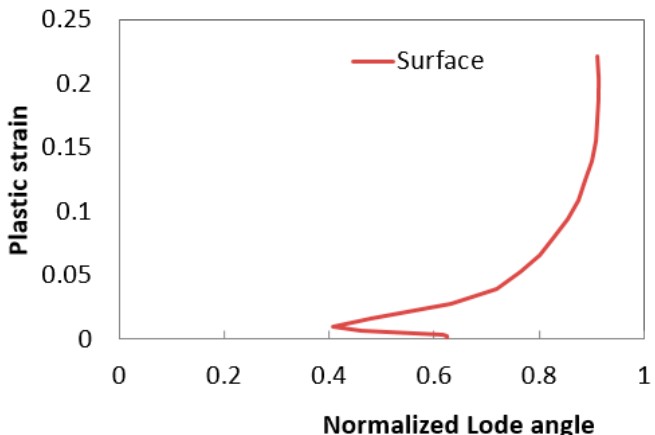

**Figure 27.** Lode angle for notched tensile test.

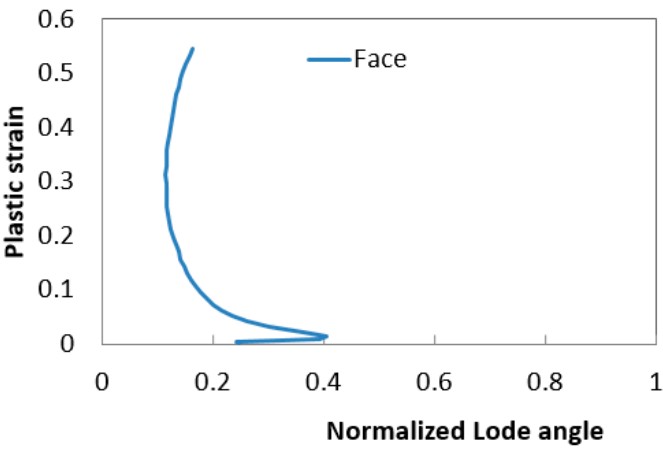

**Figure 28.** Lode angle for shear test.

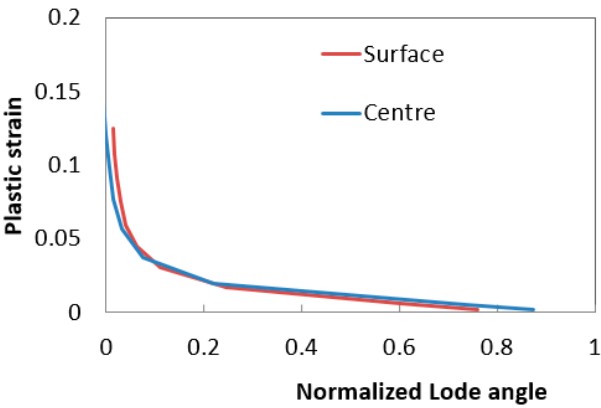

**Figure 29.** Lode angle for plane strain test.

The Oyane fracture criterion is based on stress triaxiality. The parameters of the Oyane criterion can be evaluated based on literature data [24]. The equation for the criterion can be rewritten as:

$$\varepsilon_f = D_C - \frac{1}{a_0} \int_0^{\varepsilon_f} \frac{\sigma_m}{\sigma_{eq}} d\varepsilon, \tag{12}$$

This expression presents a linear relationship between the fracture strain and the integral:

$$\int_0^{\varepsilon_f} \frac{\sigma_m}{\sigma_{eq}} d\varepsilon, \tag{13}$$

which can be obtained from numerical simulation. The graphic representation of Equation (12) is shown in Figure 30. The material constants for the Oyane criterion can be determined from a linear fit to points for the individual tests. Material constant $a_0$ can be derived from the slope of the line as its negative inverse value. *Dc* is in fact the value at the intersection of the line with the ordinate. The values determined in this manner were *Dc* = 0.63 and $a_0$ = 0.26.

When Lode angle is considered for the Oyane criterion, see Figures 27–29, the notched tensile test becomes the least relevant of the three tests. Lode angle for the notched tensile test is far from the other two tests. It seems appropriate to have the values of Lode angle close to each other, see the dependence illustrated in Figure 1. In an alternative course of calculation, the notched tensile test was omitted from the evaluation of material constants. The resulting values *Dc* = 0.65 and $a_0$ = 0.27 do not differ substantially from the previous pair of values.

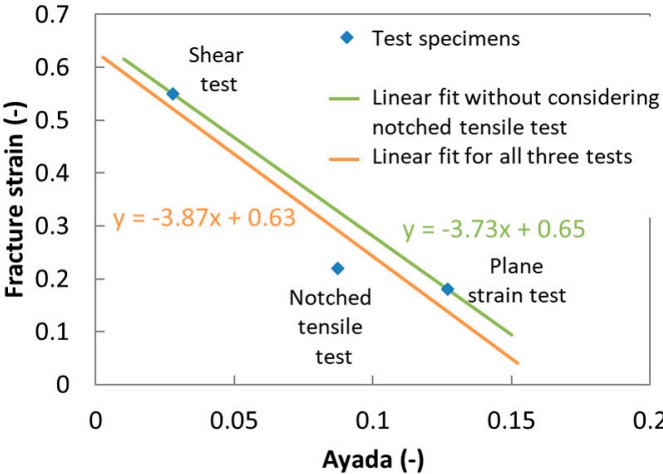

**Figure 30.** Relationship between $\int_0^{\varepsilon_f} \frac{\sigma_m}{\sigma_{eq}} d\varepsilon$ and $\varepsilon_f$.

Figure 31 shows a graphic representation of the Oyane criterion along with the values from Table 4. in a plot of fracture strain versus triaxiality.

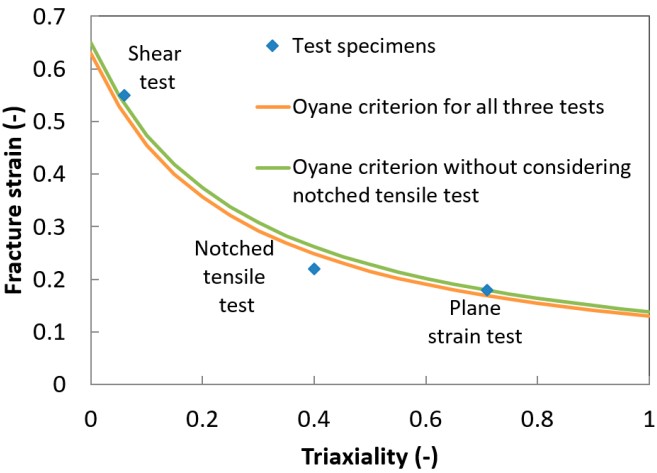

**Figure 31.** Graphic representation of the Oyane criterion.

The damage coefficients for the Oyane criterion are summarized in Table 5.

**Table 5.** Critical damage values for the Oyane criterion.

| Specimen Type | Critical Damage $D_C$ (-) | Material Coefficient $a_0$ (-) |
|---|---|---|
| All specimens | 0.63 | 0.26 |
| Shear and plane strain only | 0.65 | 0.27 |

## 6. Validation of Criteria Coefficients and Discussion

The main purpose of numerical analysis of damage is to predict the failure location and time in a part under load. The validity of the above-calculated critical damage values and material constants can be assessed by comparing numerical simulations with experimental data. This validation was based on the third group of experimental tests: notched and unnotched ring specimens and notched cylinder specimens. All these tests were simulated using NCL and Oyane fracture criteria with accurate replication of the test conditions.

### 6.1. Radial Compression of Ring Specimens

Two types of ring specimens were tested: notched and unnotched ones. The primary purpose of the notches was to ensure the ring fractures. However, once a simulation of compression of an unnotched ring proved that it would suffer fracture as well, an additional actual test of an unnotched ring was performed to validate the simulation. The simulation models are shown in Figures 32 and 33.

As expected, the numerical analysis confirmed that fracture in the notched ring starts at the notch tip. The fracture propagates inside the ring with the increasing load, opposite the line of contact with the dies. In the unnotched ring, fracture initiates first inside the ring, opposite the outer line of contact with the dies. Then, as loading continues, another fracture occurs on the outer surface where the ring bulges out. Fracture propagation was represented by the softening method implemented in the DEFORM code.

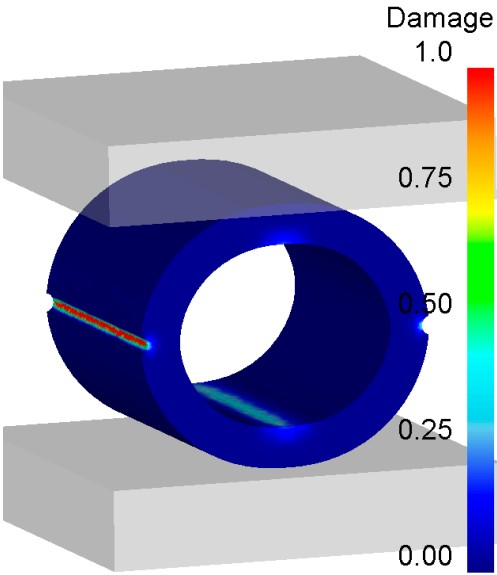

**Figure 32.** Simulation model of compression of a notched ring.

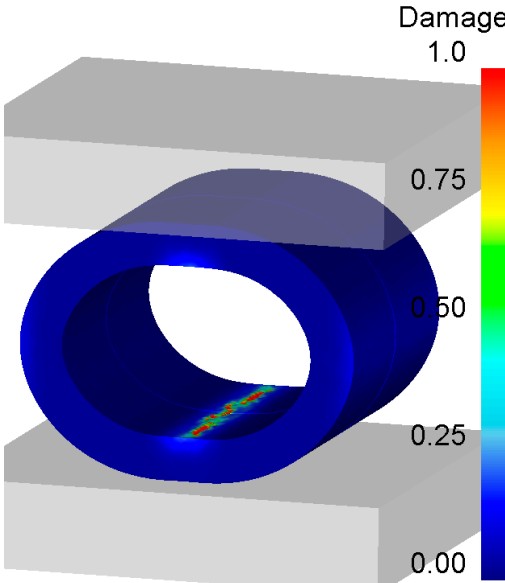

**Figure 33.** Simulation model of axial compression of an unnotched ring.

The simulation model with the NCL criterion had two versions, each for one of the two critical damage values listed in Table 4. The forces from these simulations and from the experiments are compared in Figure 34. The calculated material constants for the Oyane criterion, as presented in Table 5, were substituted into simulation and the resulting simulated forces were compared with the measurement data, as shown in Figure 35.

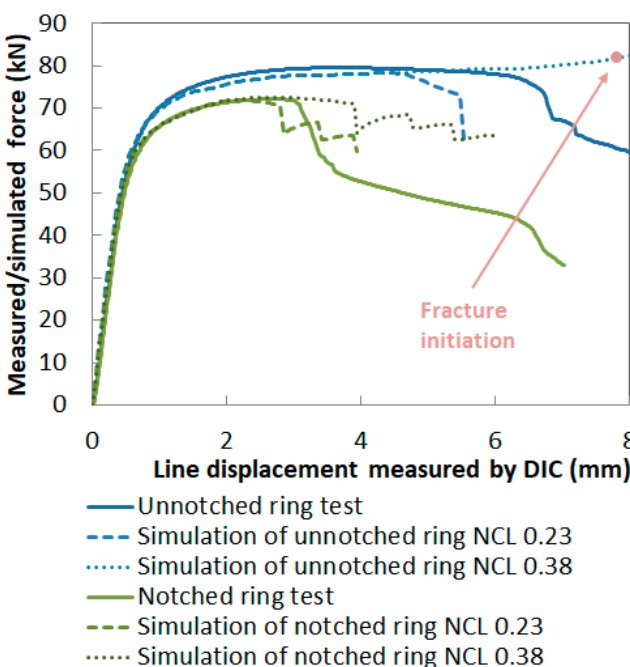

**Figure 34.** Measured forces in radial compression of rings and the forces simulated with the use of the NCL fracture criterion.

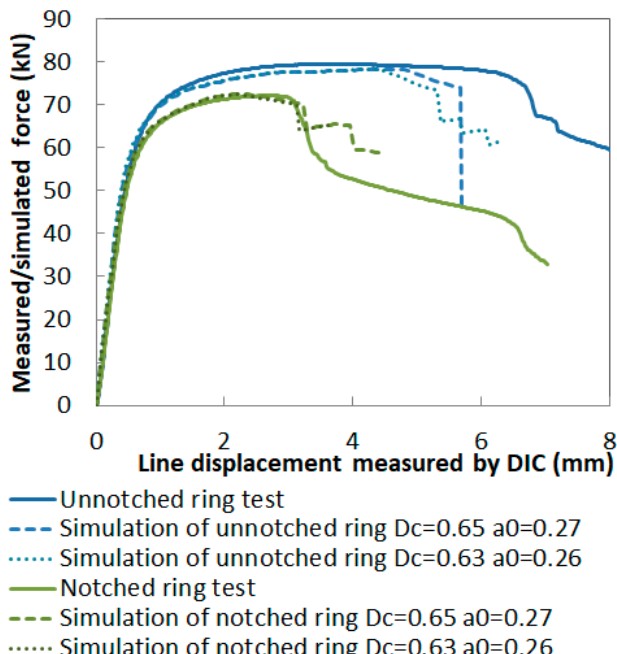

**Figure 35.** Measured forces for notched and unnotched rings and the forces simulated with the use of the Oyane fracture criterion.

The location of the fracture initiation was predicted correctly by simulation with both the NCL and Oyane fracture criteria. For the notched ring test and the NCL criterion, the simulation model with the value of 0.23 predicted a premature fracture, whereas the one with the value of 0.38 predicted failure later than in the experiment. The actual unnotched ring test and its simulation yielded similar results. However, no drop occurred on the simulated force curve for the critical damage value of 0.38. In the simulations, the time of fracture initiation was understood to be the moment the stress decreased, as indicated by the softening method. The point at which the stress begins to fall is indicated by

the red arrow in Figure 34. It illustrates an important finding that although a drop in the simulated force demonstrates that fracture is present, the actual crack initiation might have occurred earlier. During these experiments, the specimen surfaces had to be carefully observed in order to detect fracture. However, the DIC camera was aimed at the specimen face to measure the displacement. The other surfaces were only observed with the naked eye. Hence, the exact time of crack initiation was not determined.

The NCL critical damage value 0.38 in Table 4 applies to the shear test. The value for the notched ring test is the same as the value for the tensile test. The more appropriate value appears to be 0.23, which predicts a slightly earlier failure. The critical damage for the NCL criterion depends significantly on the stress state. This criterion should therefore be used with care.

The simulation of notched ring compression with the Oyane criterion showed very good agreement with the experiment on the notched ring. However, the failure prediction for the unnotched ring was earlier than in the actual test. The calculations with different coefficients gave almost identical results, indicating that the dependence on Lode angle has of little importance for this material with respect to the used triaxiality range from around 0 to 0.7, see Table 4. This result was predictable due to the small difference in the coefficients as shown in graphical representation of Oyane criterion in Figure 31.

*6.2. Axial Compression of a Notched Cylinder*

Axial compression of a notched cylinder was also used for validating both criteria (Figure 36). The measured and simulated forces for the NCL and Oyane criteria are plotted and compared in Figure 37. In this test, failure is not indicated by a drop in the test force. In the actual tests, the moment of fracture initiation was found by means of DIC measurement. In simulations, the fracture propagation is represented by the softening method. Hence, the crack initiation is identified as a decrease in stress in the relevant region.

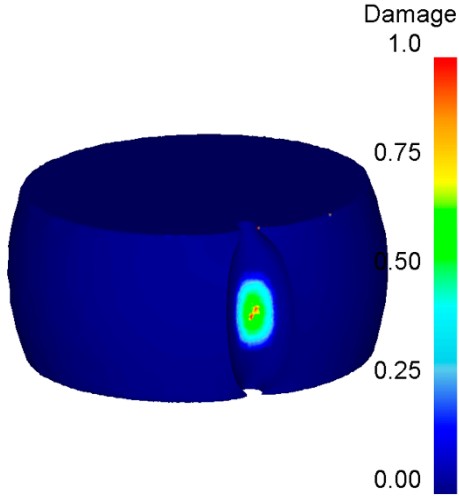

**Figure 36.** Simulation model of notched cylinder compression.

In the notched cylinder specimen, the NCL criterion predicted cracking rather early for $Dc = 0.23$. The simulation with $Dc = 0.38$ shows better agreement with the actual test data. With reference to the critical values in Table 4, the NCL value from the shear test ($Dc = 0.38$) appears to better predict fracture in this shear dominated process.

Fracture predictions for notched cylinder obtained with the use of Oyane criterion were very accurate, with the fracture being located in the centre of the range of measured fracture displacement values. The simulation with Oyane criterion was performed for two different coefficients, see Figure 31, that were chosen to assess the influence of the Lode angle. As expected, the difference is small for this material and the used triaxiality range, see Table 4.

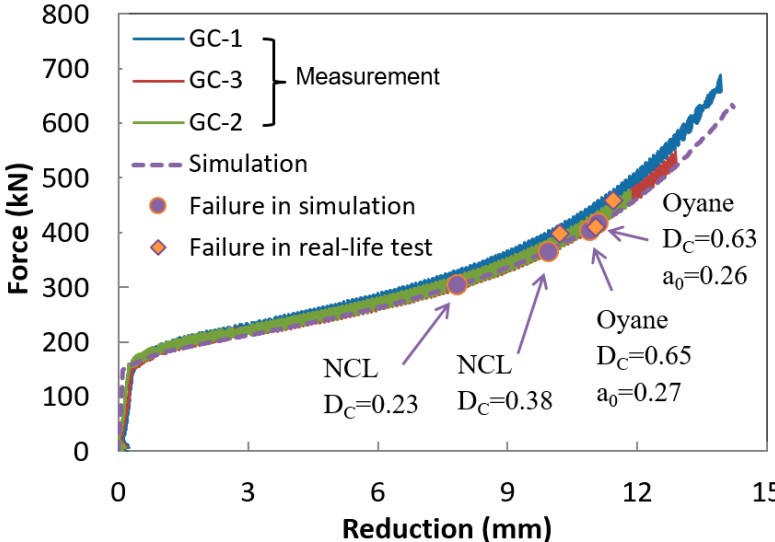

**Figure 37.** The comparison between measured and simulated compression forces and fracture initiation points for notched cylinder specimens.

Both validation tests showed quite a good applicability of the Oyane criterion, based on fracture parameters from the simple tests performed in this study. The Oyane criterion covers a wider range of stress states and industrial problems using two inserted parameters, as reflects the dependence on stress triaxiality. The NCL model can also predict failure rather accurately but the choice of the critical damage value is strongly dependent on the stress state in the process. The failure prediction in radial compression of rings was more accurate with $Dc = 0.23$, whereas the simulations of axial compression of notched cylinders gave better results for $Dc = 0.38$. In the rings, tensile stress dominated in fracture initiation locations. By contrast, failure in the notched cylinders was shear dominated. Therefore, a different critical damage value was needed for a reliable failure prediction.

## 7. Conclusions

Tensile and compression tests were conducted to determine plasticity data for DIN 34CrNiMo6 steel. Notched tensile tests, shear tests and plane strain tests were used for finding critical damage values for the Oyane and NCL criteria. Fracture prediction simulations were carried out using DEFORM code. The choice of tests for sufficiently precise fracture prediction was dictated from an engineering point of view. Plasticity and fracture models were validated using simple tests: radial compression of notched and unnotched rings and axial compression of notched cylinders.

Reducing the inaccuracy of the material flow stress is an important issue for increasing accuracy in subsequent predicting the fracture behaviour. Both fracture criteria predicted the failure locations accurately in the validation tests. The time of fracture was predicted more accurately by the Oyane criterion for both tests. The Oyane criterion can be used for failure prediction in industrial problems without any extra experience of and knowledge with measuring the data for material models.

The NCL criterion lacks general applicability, unlike the Oyane criterion. The effect of both triaxiality and Lode angle in the NCL criterion is hidden in maximal principal stress. The NCL criterion does not allow a damage prediction for arbitrary processes but only for similar processes in which the stress state of material does not differ significantly for the critical damage value. The results indicated that it is essential to identify an applicability range for each critical damage value selected for the NCL criterion.

This paper can serve as a guide for fracture evaluation based on a limited number of simple tests. The effectiveness of this procedure was demonstrated above. Using the fracture criteria available in the DEFORM software, one can obtain the desired results without any programming tasks.

Follow-on investigations will involve validating both fracture criteria for other types of specimens. In addition, other fracture criteria will be examined in order to improve failure prediction in industrial conditions.

**Author Contributions:** Data curation, M.R.; funding acquisition, M.Z.; investigation, I.P.; methodology, M.Z. and J.D.; project administration, J.D.; writing—original draft, I.P. All authors have read and agreed to the published version of the manuscript.

**Funding:** European Regional Development Fund: CZ.02.1.01/0.0/0.0/17_048/0007350. European Regional Development Fund: CZ.02.1.01/0.0/0.0/16_019/0000836.

**Acknowledgments:** The paper was developed with a support from the ERDF project Pre-Application Research of Functionally Graduated Materials by Additive Technologies, No. CZ.02.1.01/0.0/0.0/17_048/0007350 and from ERDF Research of advanced steels with unique properties, No. CZ.02.1.01/0.0/0.0/16_019/0000836.

**Conflicts of Interest:** The authors declare no conflict of interest.

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
