# Peer review of "Using DEFORM Software for Determination of Parameters for Two Fracture Criteria on DIN 34CrNiMo6"

_metals, doi:10.3390/met10040445_

Round 1

Reviewer 1 Report

General comments

This work centers on applying two different fracture criteria, NCL and Oyane criteria in simulation software to predicting fractures in a martensitic steel. The simulation work made use of parameters derived from primary experiments that are normally standard and are documented in many test standards.  

I do follow the clarity of the computer simulation in predicting the fracture behaviour and their validity by experimental data, and I think this is the focus of the work. I found it a bit convoluted that there as much narrative about standardized test methods to establish or determine the parameters used in the simulation. Reports of tensile, compression, notched and shear tests that are standard is an unnecessary detour from the main message. It is enough to quote that these parameters are obtained from these tests as described by a referenced standard. The discussion of such primary data as expressed in Figures 16-20, takes focus away from the real message of this work. Sections 4.1, 4.2, 5.1 and 5.2 could be considerably shortened since they are not even captured (and rightfully so) in the final conclusions. 

Title:

The title of this work is not an adequate description of the final message which is about a computer simulation of fracturing in DIN 34CrNiMo6 steel using two fracture criteria. I would like to see the title reconstructed to capture the main message. 

Abstract:

I would like to see some context about the applicability and the need for this kind of outcome from the computational mechanics described here. For example, how would this informed the forming and shaping into tools and devices of these grades of steel?   The statement: "The aim of this study was to assess the applicability of fracture criteria implemented in a commercial code to general engineering applications" needs to be revised. 

Line 109  "Many published experimental reports(e.g[refs]..." should include references. 

Line 125  "4.1. Basic tests"   Tensile and compression tests? I really do not see much need for reporting this. 

Line 136: "...implemented in GOM Aramis software..."  Please provide a reference or give a more detailed narration of this method. 

Line 140  "...with red crosses in the figure above." should be "with red crosses as indicated in Figure 3."

Line 144  "4.2. Advanced tests"    Notched fracture tests? Again these are standard tests and need not be reported as detailed. 

Line 161 "4.3. Validation tests"  Experimental evaluation of fractures in compressed geometries? Calling this validation test is putting the cart before the horse! Yes, they validate (not all the way), only after it is compared to simulations. What if the outcome comparison of experimental data with simulation does not agree would it still qualify as a validity test? 

Line 179: "This technique was reported in a number of papers 179 [18]." should be:

"This technique was reported by several authors (e.g. ref[18])."

Line 235: include reference as: "because numerous researchers suggest [ref]..."

Line 266 "6. Validation of criteria coefficients and discussion" Comparing experimental and simulated fracture prediction? 

Author Response

Response to Reviewer 1 Comments

Point 1: I do follow the clarity of the computer simulation in predicting the fracture behaviour and their validity by experimental data, and I think this is the focus of the work. I found it a bit convoluted that there as much narrative about standardized test methods to establish or determine the parameters used in the simulation. Reports of tensile, compression, notched and shear tests that are standard is an unnecessary detour from the main message. It is enough to quote that these parameters are obtained from these tests as described by a referenced standard. The discussion of such primary data as expressed in Figures 16-20, takes focus away from the real message of this work. Sections 4.1, 4.2, 5.1 and 5.2 could be considerably shortened since they are not even captured (and rightfully so) in the final conclusions. 

Response 1: I used all the standard test in this paper for description of the whole procedure for obtaining the proper hardening curve and fracture coefficients. Graphs 16-20 show how the measured test on the basis of calculated flow stress data fits on all specimens used.

Point 2: Title:

The title of this work is not an adequate description of the final message which is about a computer simulation of fracturing in DIN 34CrNiMo6 steel using two fracture criteria. I would like to see the title reconstructed to capture the main message. 

Response 2: I changed the title from “Material models and determination of fracture parameters for FEM simulation using DEFORM software“ to „ Using DEFORM software for determination of parameters for two fracture criteria on DIN 34CrNiMo6 “

Point 3: Abstract:

I would like to see some context about the applicability and the need for this kind of outcome from the computational mechanics described here. For example, how would this informed the forming and shaping into tools and devices of these grades of steel?   The statement: "The aim of this study was to assess the applicability of fracture criteria implemented in a commercial code to general engineering applications" needs to be revised. 

Response 3: I changed "The aim of this study was to assess the applicability of fracture criteria implemented in a commercial code to general engineering applications" to “The aim of this study was to calibrate material model with two fracture criteria available in DEFORM software on DIN 34CrNiMo6“

I also added into abstract, row 13: “These tests are used in engineering applications for ease of manufacturing and strong ability to fracture.“

Point 4: Line 109  "Many published experimental reports(e.g[refs]..." should include references. 

Response 4: line 94: Reference on Bai – Wierzbicki study [17] was added.

Point 5: Line 125  "4.1. Basic tests"   Tensile and compression tests? I really do not see much need for reporting this. 

Response 5: The chapter is part of whole procedure to the description of hardening curve creation. Fracture data are related to properly calculated flow stress data as the damage criterion in uncoupled.

Point 6: Line 136: "...implemented in GOM Aramis software..."  Please provide a reference or give a more detailed narration of this method. 

Response 6: line 141: There is no user manual so I added reference on html pages of this software.

Point 7: Line 140  "...with red crosses in the figure above." should be "with red crosses as indicated in Figure 3."

Response 7: The sentence was corrected according your proposal.

Point 8: Line 144  "4.2. Advanced tests"    Notched fracture tests? Again these are standard tests and need not be reported as detailed. 

Response 8: line 148: “Advanced test” was wrong title used, I changed it on “Tests for fracture coefficients determination“. These tests are part of whole procedure of setting material model with fracture parameters.

Point 9: Line 161 "4.3. Validation tests"  Experimental evaluation of fractures in compressed geometries? Calling this validation test is putting the cart before the horse! Yes, they validate (not all the way), only after it is compared to simulations. What if the outcome comparison of experimental data with simulation does not agree would it still qualify as a validity test? 

Response 9: The validation specimens are compressed but there is tensile stress dominated fracture in rings and shear stress fracture in notched cylinder.

If the measured values did not agree with the simulation, then the whole described procedure would be wrong.

Point 10: Line 179: "This technique was reported in a number of papers 179 [18]." should be: "This technique was reported by several authors (e.g. ref[18])."

Response 10: line186: There is noted only one literature so the sentence was corrected according your proposal.

Point 11: Line 235: include reference as: "because numerous researchers suggest [ref]..."

Response 11: line 242: I added the reference on Bai – Wierzbicki study [17].

Point 12: Line 266 "6. Validation of criteria coefficients and discussion" Comparing experimental and simulated fracture prediction? 

Response 12: It is comparison of the experiment and simulation which shows validation of used coefficients in two different damage criteria.

Reviewer 2 Report

Dear authors,

The paper shows the ductile fracture analyses by FE simulation, and it was compared to the results of a variety of mechanical tests. Two fracture criteria, Cockroft-Latham and Oyane, including commercial FE code(Deform), were validated by your works.

The study is useful for industrial works. I think that the manuscript is worth to publish with the following mandatory revisions:

1) Is the "Deform" essential for your study?? It is a one of commercial FE code, and the other solvers will provide the same results. Please generalize the title and introduction about the expression of FE code.

2)Introduction: Tasks for conventional CAE analyses, detail of your purpose, and originality of your study should be described. Description from a industrial aspects will enrich your introduction to hit the readers.

3) Figure 1: Please check the copyright of this figure. 

4) Uncoupled fracture models: Cockcroft-Latham can be expressed by Lode-angle and stress triaxiality. From the former of this section, Cockcroft-Latham should be decompose these two components. Since Oyane include only stress triaxiality component, the comparison of two criteria will give the dependency of Lode-angle on fracture strains. 

5) Construction of plasticity models: Yield stress, tensile strength, and uniform elongation should be presented. These are essential information of the material work hardening.

6) Identification of fracture parameters: For all the mechanical tests(experiments and simulations), how did you evaluate stress and strain at the fracture portion? The distribution of the state values are not uniform, thus, the region for the evaluation is very important. The detail of these evaluation technique should be explained with figures.

7) Identification of fracture parameters: I did not understand why Asada's criterion is explained. Both Oyane and Cockcroft-Latham also include stress triaxiality. The effect of Lode-angle can be elucidate by comparison of these two criteria. The construction of this part should be revised to clarify the effect of Lode-angle.

8) Validation of criteria coefficients and discussion: The revision in 6) and 7) have to reflect in this section.

Sincerely

Author Response

Response to Reviewer 2 Comments

Point 1: Is the "Deform" essential for your study?? It is a one of commercial FE code, and the other solvers will provide the same results. Please generalize the title and introduction about the expression of FE code.

Response 1: All the simulations in this paper was done using DEFORM software, so I thought it appropriate to include the software name in the title. DEFORM is extremely effective software in a wide range industrial forming applications, research and development. The idea was to test and validate material with tools implemented in DEFORM software used by practitioners without any further programming.

Point 2: Introduction: Tasks for conventional CAE analyses, detail of your purpose, and originality of your study should be described. Description from a industrial aspects will enrich your introduction to hit the readers.

Response 2: I added in abstract - row 10: The purpose is in proposal of type of simple tests, which are sufficient for determination of damage parameters.

row 13: These tests are used in engineering applications for ease of manufacturing and strong ability to fracture.

Point 3: Figure 1: Please check the copyright of this figure. 

Response 3: Figure 1 used from literature [9] was replaced by own created graph.

Point 4: Uncoupled fracture models: Cockcroft-Latham can be expressed by Lode-angle and stress triaxiality. From the former of this section, Cockcroft-Latham should be decompose these two components. Since Oyane include only stress triaxiality component, the comparison of two criteria will give the dependency of Lode-angle on fracture strains. 

Response 4: Fracture behaviour of material is in general dependent on Lode-angle and stress triaxiality. The equation of Normalized Cockcroft-Latham criterion in DEFORM software is not dependent on these parameters. It is necessary to know the stress state of the material for the correct use of NCL, so this is the reason that it should be used with care. On the other hand the Oyane criterion has stress triaxiality in its equation.

The effect of Lode-angle on fracture strain seems small for this material, see different equation for Oyane criterion in Figure 31, where was omitted notched tensile test with different Lode angle from the other two tests with the similar values.

Point 5: Construction of plasticity models: Yield stress, tensile strength, and uniform elongation should be presented. These are essential information of the material work hardening.

Response 5: Table 2 was added with basic measured mechanical data.

Point 6: Identification of fracture parameters: For all the mechanical tests (experiments and simulations), how did you evaluate stress and strain at the fracture portion? The distribution of the state values are not uniform, thus, the region for the evaluation is very important. The detail of these evaluation technique should be explained with figures.

Response 6: Fracture strain is gained from simulation of real process in the time when the force-displacement curve begins to drop, see chapter 5.2. The location of the fracture strain or damage is seen on figures 21-23.

Point 7: Identification of fracture parameters: I did not understand why Ayada's criterion is explained. Both Oyane and Cockcroft-Latham also include stress triaxiality. The effect of Lode-angle can be elucidate by comparison of these two criteria. The construction of this part should be revised to clarify the effect of Lode-angle.

Response 7: Ayada criterion only helps to find the coefficients for Oyane criterion, because of linearity in equation (11) – it is easy to get values from DEFORM software without complicated calculation. I understand that a mention of Ayada criterion is a bit misleading here, so I deleted the reference to it.

The effect of Lode angle can not be elucidate by comparison of both used criteria by reason that none of them has its equation dependent on Lode angle.

Point 8: Validation of criteria coefficients and discussion: The revision in 6) and 7) have to reflect in this section.

Response 8: I wrote additional evaluation to the chapter 6 about the dependence on Lode angle in the limits used in this work.

row 314 (ring specimen): The calculations with different coefficients gave almost identical results, indicating that the dependence on Lode angle has of little importance for this material with respect to the used triaxiality range from around 0 to 0.7, see Table 4. This result was predictable due to the small difference in the coefficients as shown in graphical representation of Oyane criterion in Figure 31.

row 333 (notched cylinder): The simulation with Oyane criterion was performed for two different coefficients, see Figure 31, that were chosen to assess the influence of the Lode angle. As expected, the difference is very small for this material and the used triaxiality range, see Table 4.

Reviewer 3 Report

The paper describes the attempt of predicting location and failure load for simple structure or components, by a commercial software incorporating ductile fracture models. Parameters for these models are tuned by experimental tests on standard steel specimens.

Despite the accuracy of predictions on simple tests, in which however it is not known the local triaxiality state, theoretical basis seem to be missing or too poor; discussion of results is only explicative of what it is found; and findings are difficult to take as general conclusions scientifically valid. For all these reasons, the paper should be accurately revised before being considered for publication.

Other minor comments can be the following: 

  • introduction, especially in the first paragraph, should need to be rewritten cause it introduces not clearly the problem;
  • section 2: it could be useful to report the nominal mechanical characteristics of the used material (Rs Rm Kc εf...);
  • fig. 1 is not well explained in all parameters involved;
  • row 97: not clear; damage initiation and evolution are two different concepts;
  • row 119: not clear if the model used incorporates the Lode angle; if not, it is difficult to understand how it can explain the fracture;
  • section 4.2: which are the considered triaxialities? no details are reported about the tests;
  • section 4.3: iti snot clear why the authors choose such geometries;
  • row 230: triaxiality value is known to change dramatically during the stress evolution, an explanation of assuming an average value must be provided
  • the discussion involving the Lode angle is not clear.

Author Response

Response to Reviewer 3 Comments

Point 1: Despite the accuracy of predictions on simple tests, in which however it is not known the local triaxiality state, theoretical basis seem to be missing or too poor; discussion of results is only explicative of what it is found; and findings are difficult to take as general conclusions scientifically valid. For all these reasons, the paper should be accurately revised before being considered for publication.

Response 1: It makes no sense to deal with the state of triaxiality in the basic tensile and compression tests, because these tests were performed only to obtain a material hardening curve. The damage parameters are defined from second group of tests (notched tensile test, shear test and plane strain test), and their triaxiality is displayed in the graphs 24-26. This triaxiality paths correspond to the location where fracture is expected to occur, and where is the biggest strain value, see figure 21-23.

Point 2: introduction, especially in the first paragraph, should need to be rewritten cause it introduces not clearly the problem

Response 2: Changed from: “Many engineering problems can be solved by using an approach involving certain simplification. Simplification leads to minor inaccuracies, but the aim of this paper was not to develop highly-accurate failure prediction calculations but to demonstrate quick and sufficiently accurate predictions based on very few – and if possible simple – experiments.”

To: “The aim of this paper is to develop a procedure for failure prediction through simulations when limited amount of tests are available. The purpose of the paper is to demonstrate quick and sufficiently accurate predictions based on very few – and if possible simple – experiments.”

Point 3: section 2: it could be useful to report the nominal mechanical characteristics of the used material (Rs Rm Kc εf...);

Response 3: Table 2 was added with basic measured mechanical data.

Point 4: fig. 1 is not well explained in all parameters involved;

Response 4: I changed figure 1 and rewrote the description in paragraf, and I added information about the black line in the graph.

Point 5: row 97: not clear; damage initiation and evolution are two different concepts;

Response 5: That’s true, I changed it to make it clearer: In DEFORM code, the growth and coalescence of voids that lead to the evolution of fracture can be modelled by either the softening method or by element deletion.

Point 6: row 119: not clear if the model used incorporates the Lode angle; if not, it is difficult to understand how it can explain the fracture;

Response 6: The paragraf was changed to: “The fracture criteria examined here do not incorporate the Lode angle as a fracture variable. However, it is useful to look at the influence of Lode angle on failure, as it was showed in Figure 1. It can be done by separating the fracture coefficients for tests with different Lode angle.”

Point 7: section 4.2: which are the considered triaxialities? no details are reported about the tests;

Response 7: Chapter 4.2 deals only with the measuring of the tests used. More information about the triaxialities of these tests is in the chapter 5.2, which deals with simulations of the tests. There is presented triaxiality (fig. 24- 26) and the location of this value on the specimens (fig. 21-23).

Point 8: section 4.3: it is not clear why the authors choose such geometries;

Response 8: I added in chapter 4.3: Based on extensive literature survey it was found that presented geometry of test specimens is easy to manufacture and it has strong ability to fracture initiation.

Point 9: row 230: triaxiality value is known to change dramatically during the stress evolution, an explanation of assuming an average value must be provided

Response 9: I added: The triaxiality is changing significantly during the stress evolution, so an average value is usually utilized and recommended.

Point 10: the discussion involving the Lode angle is not clear.

Response 10: I added in row 314: The calculations with different coefficients gave almost identical results, indicating that the dependence on Lode angle has of little importance for this material with respect to the used triaxiality range from around 0 to 0.7, see Table 4. This result was predictable due to the small difference in the coefficients as shown in graphical representation of Oyane criterion in Figure 31.

And to row 333: The simulation with Oyane criterion was performed for two different coefficients, see Figure 31, that were chosen to assess the influence of the Lode angle. As expected, the difference is very small for this material and the used triaxiality range, see Table 4.

Round 2

Reviewer 2 Report

Dear authors,

I confirmed the improvement of your manuscript. However, I recommend further revision as follows: 

1)Table 2: Please do not use  abbreviations. The manuscript have to be easily read for a variety of readers.

2) Response 4: Your response is wrong. Principal stress is expressed as

sigma_I~=equivalent_stress(stress_triaxiality+2/3 cos(pi/6 (1-Lode_angle)).

It is well-known expression about maximum principal stress. Thus, Normalized Cockcroft-Latham criterion is dependent on Lode angle. It is also well known. I required correction of the explanation about relation between Cockcrof-Latham and Lode angle in all parts of your manuscript in order to avoid misleading the readers.  

Sincerely

Author Response

Response to Reviewer Comments

Point 1: Table 2: Please do not use  abbreviations. The manuscript have to be easily read for a variety of readers.

Response 1: I changed the description in the table, no abbreviations are used now.

Point 2: Response 4: Your response is wrong. Principal stress is expressed as

sigma_I~=equivalent_stress(stress_triaxiality+2/3 cos(pi/6 (1-Lode_angle)).

It is well-known expression about maximum principal stress. Thus, Normalized Cockcroft-Latham criterion is dependent on Lode angle. It is also well known. I required correction of the explanation about relation between Cockcrof-Latham and Lode angle in all parts of your manuscript in order to avoid misleading the readers.  

Response 2: Thank you for your remark, I am very grateful for the comment, I did not realize this dependence. I removed any description where the Lode angle dependence was denied.

I added:

Row 82: Nevertheless, the NCL criterion is based on maximal principal stress, which is dependent on both the triaxiality and the Lode angle. The NCL criterion only averages the effect of both parameters using maximal principal stress.

Row 85: The Oyane criterion includes only stress triaxiality dependence. Oyane prefered triaxiality, which seems to be more significant for crack initiation and evolution.

Row 117: The maximal principal stress can be expressed by stress triaxiality and Lode angle [18,19]:

where η is triaxiality and is a normalized Lode angle.

Row 126: Only the NCL criterion has a built-in dependence on the Lode angle. The effect of the Lode angle on Oyane criterion can be done by separating the fracture coefficients for tests with different Lode angle.

Row 231: This time instant and location indicate the critical damage value for the NCL criterion which is characterized by normalized Lode angle and triaxiality.

Row 340: The Oyane criterion covers a wider range of stress states and industrial problems using two inserted parameters, as reflects the dependence on stress triaxiality. The NCL model can also predict failure rather accurately but the choice of the critical damage value is strongly dependent on the stress state in the process.

Row 360: The effect of both triaxiality and Lode angle in the NCL criterion is hidden in maximal principal stress. The NCL criterion does not allow a damage prediction for arbitrary processes but only for similar processes in which the stress state of material does not differ significantly for the critical damage value.

Reviewer 3 Report

The authors did accomplish all the tasks I suggested, and answered clearly to my doubts about the method. I think the paper can be published now.

Author Response

Thank you very much for your comment.

I.Polakova

Round 3

Reviewer 2 Report

The manuscript was improved to be published.

This manuscript is a resubmission of an earlier submission. The following is a list of the peer review reports and author responses from that submission.

Round 1

Reviewer 1 Report

The authors gave an excellent theoretical background that provides a basis for their modeling.  I like the fact that even fundamental experiments were done to accompany the final modeling. It makes the paper complete to a reader. 

I will like the authors to add to the abstract the final findings captured in the conclusion. It is essential to capture the modeling outcome so that readers starting wit abstract would know what to expect. It will also be a good abstract to place these findings in the context of current understanding in the literature on the fracture mechanics of this type of material.

Reviewer 2 Report

Dear authors,

the manuscript presented reviews of two types of ductile fracture criteria bundled in Deform. Plural types of mechanical tests were conducted for validation of Cockcroft-Latham and Oyane criteria. The revies suggested high applicability of Oyane criterion.

I understand the usefulness of such reports, but it is not suitable for scientific paper. In a view of scientific paper, originality of this manuscript is not high. Generality of the information is not also high because the review is only for specific software, “Deform”. Additionally, the manuscript does not include discussion. Readers cannot comprehend why the FE simulation resulted in the high applicability of Oyane criterion.

For the structure of the manuscript, method, result, and discussion have to be classified. They should be written in different sections.

I think that detailing the mechanism must remarkably improve this manuscript, and it is mandatory for publication.

Sincerely  

Reviewer 3 Report

The aim of the present study was to test uncoupled fracture models (plasticity independent) available in DEFORM software. The authors attempted to use experimental plasticity data (Part 4) to ‘calibrate’ the critical damage value in the NCL and Oyane criterion (Part 5). However, uncoupled fracture criteria have principally no effect on material behavior prior to final fracture and can be treated separately from plasticity models [V.V. Vershinin, Int. J. Solids Struct. 67-68 (2015) 127-138]. As a result, this kind of ‘calibration’ could be invalid at the beginning.

It is recommended that this kind of ‘calibration’ should be applied to a coupled fracture criteria, for instance, the classic GTN, which is definitely not ‘computationally expensive’. The present calculation was almost linear and far too simple.

Additional comments are listed below:

Generally:

The language in the introduction part should be improved in comparison to the rest part of the work.

Page 1 line 24:

The data, however, depend on process variables, predominantly temperature and strain rate.

Not the date, but the Strain hardening and fracture behavior depend on …

Page 1 line 29:

continue attempting to predict

Page 1, Line 29 to 39:

The review of classic works could be better, as in [Y.L. Bai, T. Wierzbicki, Eng. Fract. Mech. 135 (2015) 147-167].

Page 6, Line 173-185:

Please do not mix the experimental and simulation parts.